# Mapping Heat Stress Vulnerability and Risk Assessment at the Neighborhood Scale to Drive Urban Adaptation Planning

**Denis Maragno** [1],*, **Michele Dalla Fontana** [2] **and Francesco Musco** [1]

1   Department of Architecture and Arts, University Iuav of Venice, Venice 30135, Italy; francesco.musco@iuav.it
2   School of Public Health, University of São Paulo, São Paulo 01246-904, Brazil; mdallafontana@usp.br
*   Correspondence: dmaragno@iuav.it

**Abstract:** Climate change is one of the most complex issues of the 21st century, and even though there is general consensus about the urgency of taking action at the city level, the planning and implementation of adaptation measures is advancing slowly. The lack of data and information to support the planning process is often mentioned as a factor hampering the adaptation processes in cities. In this paper, we developed and tested a methodology for heat stress vulnerability and risk assessment at the neighborhood scale to support designers, planners, and decision makers in developing and implementing adaptation strategies and measures at the local level. The methodology combines high-resolution spatial information and crowdsourcing geospatial data to develop sensitivity, adaptive capacity, vulnerability, exposure, and risk indicators. The methodology is then tested on the urban fabric of the city of Padova, Italy. Our results show that different vulnerability and risk values correspond to different typologies of urban areas. Furthermore, the possibility of combining high-resolution information provided by the indicators and land use categories is of great importance to support the adaptation planning process. We also argue that the methodology is flexible enough to be applied in different contexts.

**Keywords:** vulnerability and risk assessment; climate adaptation planning; climate change

---

## 1. Introduction

Climate change is one of the most complex issues of the 21st century, and the magnitude of the problem is globally recognized and largely discussed both in the academic and political arena. The 14th edition of the Global Risks Report [1] assess different global risks and their potential impact. According to this analysis, the "failure of climate-change mitigation and adaptation" and "extreme weather events" (also partly associated to climate change) are the primary risks if you look at their likelihood and their potential impact at a global scale.

According to the Intergovernmental Panel on Climate Change (IPCC) [2], further warming and changes of the climate system will increase the likelihood of severe impacts for people and ecosystems. However, scholars claim that, in terms of direct human life and economic losses, the worst consequences will occur in cities [3–6]. Indeed, the increasing number and intensity of extreme weather events, changes in rainfall patterns, flooding, sea-level rise, and heat waves [7] particularly threaten urban populations, economic activities, and infrastructures in urban settlements.

Future heat waves, like those that hit Western Europe in 2003 and Eastern Europe in 2010 [8], will become more intense, more frequent, and longer lasting in the second half of the 21st century [9–11]. Due to their great agglomeration of infrastructures, people, and buildings, cities are considered particularly vulnerable to climate risks and extreme heat events [11,12] Furthermore, heat waves are

intensified in urban agglomeration by the development of urban heat islands [13] that make cities warmer than the surrounding rural and natural areas [14,15].

While different cities across Europe saw their highest recorded temperatures during the summer of 2019, the planning and implementation of adaptation measures at the local level is advancing slowly. In the last two decades, many cities incorporated mitigation measures in their policy objectives, or they have adopted mitigation strategies. The efforts to achieve greenhouse gas reduction carried out by cities worldwide (either individually or in network) are an example of bottom-up initiatives to cope with climate change. However, impacts, vulnerability, and adaptation to climate change have received less attention than mitigation at the city level [16]. Despite the progress achieved in the course of the last few years (see for example the Mayors Adapt initiative), there are still fewer examples of adaptation initiatives at the local level than mitigation initiatives [17].

To make cities "climate proof" requires a substantial modification of how they are planned, designed and managed. That includes both the strategies to reduce climate change emissions, and to make the urban systems more resilient to climate changes effects [18]. As Biesbroek et al. [5] pointed out, "both mitigation and adaptation have a spatial dimension" and coping with climate change in cities requires changes in socio-economic practices, the planning and the design of spatial and physical factors [19]. Although cities have an important role in mitigation strategies [20], the local dimension become even more relevant when it comes to adaptation. As "the geometry, spacing and orientation of buildings and outdoor spaces strongly influence the microclimate in the city" [21], urban planning and design contributions are fundamental to face increasing temperatures and other climate change effects.

We argue that local authorities, along with urban planners and designers, are well-placed to face the challenge. However, such a challenge requires data and information that are not always available at the level of detail needed. The lack of high-resolution and spatial information can hamper planners and designers to define suitable adaptation measures and can discourage local authorities from implementing such measures [22,23].

Uncertainty of climate models and the request for "high-resolution downscaled climate projections" are often considered as factors that are hindering progress in planning the adaptation of cities to heat waves [24,25]. Despite our acknowledging the role of climate projections in adaptation strategies, we also agree with scholars arguing that the development and implementation of adaptation strategies should not be delayed for an optimal level of prediction accuracy [24–26]. This is particularly true at the local level, where climate projections are often not able to detect differences between nearby cities, let alone informing local authorities about different climate scenarios at the neighborhood level.

In the literature, there are different cases of vulnerability assessment conducted at the city level (e.g., references [6,23–29]) that, considering the whole city as unit of analysis, allow us to compare cities that are differently susceptible to climate change risks. However, this gives little guidance to local authorities, planners, and designers of where actions are more urgently needed within the city and what kind of interventions are required. Other scholars have focused on social vulnerability and conducted more refined analysis to identify differences between areas of the same city (e.g., references [23,29]). However, these analyses are mainly based on socio-economic and demographic data that do not consider properties of surface structure and surface cover that can exacerbate climate change effects.

This study aims to fill this gap by developing a methodology to perform vulnerability and risk assessment (related to heat waves impacts) at the neighborhood scale, also taking into account properties of surface structure and surface cover into the analysis. The intent is to support local adaptation by better understanding the local territorial characteristics that contribute to increase vulnerability and risk.

More fine-scale information is necessary to produce more detailed analysis within cities and to give more specific indications for interventions. New Technologies (NTs), and particularly Information and Communication Technologies (ICTs), become extremely relevant to produce, manage and make use of spatial information [30]. Remote Sensing Technologies, for example, are very convenient to produce information that can support climate-proof planning processes. In fact, a major obstacle in

the vulnerability assessment phase (preparatory to the design of tailored responses) is the generally inadequate knowledge available to feed into the planning process. Information such as vegetation coverage ($m^2$), trees height, incident solar radiation, waterproof surfaces coverage (on public and private ground) are hardly available at the neighborhood scale. Moreover, ICTs can support the generation and management of new information. In this sense, public administrations might find it advantageous to move beyond the idea of owning the information and instead adopt a new model in which resources and knowledge are shared based on cooperation, participation, and the involvement of citizens, academia, and the private sector. One of the first step, if not the first, to the climate proof planning is to build a reliable knowledge framework that should be innovative (for the detail of the information), shared, and integrated.

In this paper, we develop and test a methodology for heat stress vulnerability and risk assessment that involves the use of high-resolution spatial information that can support city planners in developing adaptation strategies at the local level regardless of the availability of reliable high-resolution climate data. The methodology is then applied to the case of the city of Padova (Northeast Italy). We argue that, considering a simplified four-steps planning process made of (i) preparatory/analysis phase, (ii) definition of strategies and measures, (iii) implementation, and (iv) monitoring, the methodology can support all the four phases.

This paper is organized as follows: the upcoming section provides information related to the methodology developed to carry out the vulnerability and risk assessment. Section 3 shows the results of the analysis conducted in the city of Padova. The methodology and the results are then discussed in Section 4. The last section offers some conclusions.

## 2. Materials and Methods

### 2.1. Clarification on the Terminology and Methodology's Outline

The methodology for climate change vulnerability and risk assessment have developed in the last decade bringing both benefits in the quality of the results but also confusion in the terminology used. The IPCC's Fifth Assessment Report—Ar5 [2] defines a new approach to vulnerability evaluation that revises the terminology previously used in the Fourth Assessment Report —Ar4 [31]. Fritzsche and colleagues [32] underline how, in Ar5, the methodology for vulnerability and risk assessment has more in common with the disaster risk reduction approach (DRR) developed by the UN Office for Disaster Risk Reduction.

The terms "vulnerability" and "exposure" are used differently in the Ar4 and the Ar5. "While AR4 uses the concepts of sensitivity and adaptive capacity [vulnerability] to describe the moderating attributes of the system, AR5 uses the concept of exposure (the presence of a system in places that could be adversely affected) and vulnerability (predisposition to be adversely affected)" ([32] p.33). Furthermore, climate-related stresses, such as heat waves, are considered as "exposure" in the Ar4 and as "hazard" in the Ar5. We therefore consider fundamental to be clear on the terminology that we use in this work. We refer to the AR5 approach that defines "hazard" as the probability of an extreme climatic event to happen and that is able to cause damage, such as loss of life or damage to infrastructures, services, and ecosystems. "Vulnerability" is defined as the inclination of a system to be negatively influenced by the hazard and it is also considered as a variable to calculate risk. Vulnerability also includes the concepts of sensitivity (the susceptibility to harm) and adaptive capacity (the capacity to cope and adapt). The third component of risk is "exposure," which is here intended to mean the presence or not of infrastructures, services, species and ecosystems, and cultural properties in the considered area that could be adversely affected by potential impacts.

The methodology proposed is tailored to the heat wave phenomenon, and the variables considered to assess vulnerability and risk are then chosen accordingly. It is also important to highlight that, instead of social vulnerability analysis, in which the population is generally the object of the analysis, we consider geographical units (artificially defined) as the entity that we analyze.

The process that supports the methodology is shown in Figure 1. The methodology is thought to be replicable for any climate hazard and city. In this paper we describe the process considering heat waves (hazard) and associated impact in urban areas. Next step consists in data collection, which is fundamental, since the final result depends on quantity and quality of spatial data available. In the case presented, Lidar data, satellite data, high-resolution orthophotos, and Open Street Map (OSM) data have been processed in addition to the available municipality spatial data. All spatial data are then migrated into the hexagonal geo database so that it is possible to calculate all the indicators selected to measure sensitivity and adaptive capacity. Furthermore, the indicators for exposure need to be defined to assess the risk. Both vulnerability and risk assessment processes can also lead to the development of vulnerability and risk maps.

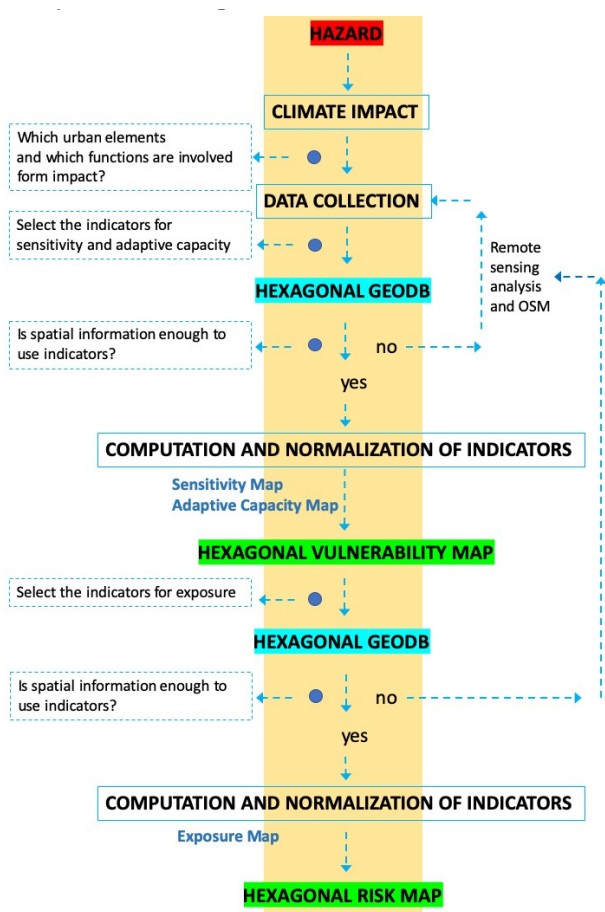

**Figure 1.** The flow chart of the methodology.

## 2.2. Presentation of the Case Study

Padua is an Italian city in the Veneto Region, situated at 40 km west of Venice, in the Po Valley. It is characterized by a humid subtropical climate, with harsh winters and hot summers, with stagnant air conditions (fog and sultry weather). The surface of the city is 93 km$^2$, and the population is approximately 210,000 (Istat, 2011). The population density of Padova is among the highest in Italy at 2300 inhabitants/km$^2$.

The city of Padova frequently suffers from Urban Heat Island effects, so much that, during the European Project "UHI-Development and application of mitigation and adaptation strategies for counteracting the global Urban Heat Island phenomenon" [33] the University Iuav of Venice and the University of Padova worked together with the Municipality of Padova to understand how UHI strikes the city.

The University of Padova made several measurement sessions during the summer of 2012. The results highlight a marked nocturnal presence of the phenomenon (3–6 °C of difference between urban and neighboring rural areas), and a lower diurnal with an average difference of 1.2–2 °C [34].

Information presented in this paper completes previous information on UHI in the context of Padova by delving deeper in the factors that exacerbate this phenomenon. The methodology presented in this paper has been used also in the European Project Life Veneto Adapt "Central Veneto Cities networking for adaptation to Climate Change in a multi-level regional perspective" (LIFE16 CCA/IT/000090), and the city of Padova is now improving its Sustainable Energy and Climate Action Plan (SECAP), also based on the results of this research. In particular, the methodology contributes to the analytical part of the plan and supports the development of strategies. Moreover, it can be applied in the monitoring stage of the SECAP.

## 2.3. Geo-Database Preparation

The database that supports the analysis is organized in a single table (entity) that is visually projected as a hexagonal grid. Each row of the table corresponds a georeferenced hexagon (with a side of 160 m) in the grid, while each column corresponds to the information and all the indicators for each hexagon. Having the information aggregated in one single table facilitates the data management and it allows the simultaneous assessment of all the information related to the portion of territory surface virtually contained in every hexagon of the grid.

The spatial complexity of cities and the need to analyze the urban fabric in great detail, made us choice to use the hexagonal grid to perform a GIS spatial analysis. The added value in employing the spatial hexagonal pattern model in territorial analysis, is the possibility of producing complex calculations in a very fast and automatic way, by using standardized units to develop the mathematical comparison of identical areas. These cells can be compared to the nearby cells or distant ones, so they provide very precise results and maximize the reading of the spatial reports. Furthermore, the hexagonal grid provides greater clarity in the visualization of the results [35].

## 2.4. Vulnerability

Vulnerability is here considered as a function of the urban area sensitivity and its adaptive capacity. The vulnerability index is calculated using the following Equation (1):

$$V = (S/n) - (AC/n),\tag{1}$$

where V is the vulnerability, S is the sensitivity, AC is the adaptive capacity, and n is the number of the indicators being used.

Sensitivity provides information about the susceptibility of cities (or territories) to specific impacts; for this reason, it is influenced by the specific properties of the system under consideration. These properties are not fixed, but they need to be consistent with the impact and to guarantee a homogeneous analysis throughout the territory. We build on references [36–40] to identify the factors that contribute to increase the susceptibility of an area to accumulate heat (sensitivity). In doing so, we consider five surface properties variables that define the sensitivity value (Table 1).

**Table 1.** Sensitivity indices used.

| Indicator/Variable | Description |
| --- | --- |
| Sky View Factor (SVF) | Ratio of the amount of sky hemisphere visible from ground level to that for an unobstructed hemisphere |
| Built Area Fraction | Ratio of building plan area to total ground area |
| Impervious Surface Fraction | Ratio of unbuilt impervious plan area (paved, sealed) to total ground area. |
| Street Incoming Solar Radiation | Potential solar radiation incoming for street surface |
| Roofs Incoming Solar Radiation | Potential solar radiation incoming for roof surface |

The adaptive capacity of an urban area is determined by its potential to adjust to heat waves, and scholars agree on the role of green infrastructure to mitigate high temperature in urban areas [12,41]. Adaptive capacity is then measured by considering two indicators—green areas and tree-cover percentage (Table 2). Remote sensing analysis of orthophotos (RGBI) with 25 cm resolution was used to identify areas covered by vegetation. Images were classified using Normalized Difference Vegetation Index (NDVI), and pixels with values greater than zero were identified. Vegetation height was obtained by interpolation of vegetation and the Digital Surface Model (DSM) that resulted from the elaboration of points cloud from LiDAR data (50 cm/pixel resolution). This resulted in an atlas of the urban vegetation classified by height.

**Table 2.** Adaptive capacity indices used.

| Indicator/Variable | Description |
| --- | --- |
| Trees | Ratio of the area covered by trees to the total ground area. |
| Green areas | Ratio of green areas (e.g., street green, green verge, house gardens, etc.) to the total ground area. |

### 2.5. Risk

Risk assessment is carried out by combining vulnerability and exposure values, considering probability of the heat wave to occur (hazard variable) as less relevant. In fact, from an urban planning perspective, vulnerability and exposure are factors that can be locally modified to some extent, while there is no control on the hazard itself.

The risk is calculated using the following Equation (2):

$$R=V*(E/n), \tag{2}$$

where R is risk, E is exposure, V is vulnerability, and n is the number of indicators being used.

Exposure to heat waves is here given by the combination of different factors that are summarized in Table 3. It is important to emphasize that, as in the case of sensitivity and adaptive capacity, also in the case of exposure the indicators have been selected according to what can be mainly affected by heat waves.

**Table 3.** Exposure indices used.

| Indicator/Variable | Description |
| --- | --- |
| Cultural heritage | Ratio of the area covered by tangible cultural heritage (e.g., historical buildings, churches, monuments, etc.) to the total ground area |
| University buildings | Ratio of the area occupied by university buildings to the total ground area |
| Industrial | Ratio of the area occupied by industrial facilities to the total ground area |
| Public buildings | Ratio of the area occupied by public buildings to the total ground area |
| Sport facilities | Ratio of the area occupied by sport facilities to the total ground area |
| Schools | Ratio of the area occupied by school facilities to the total ground area. |
| Parking lots | Ratio of the area occupied by parking lots to the total ground area |
| Bar | Number of bar within the area |
| Café | Number of café within the area |
| Restaurants | Number of restaurants within the area |
| Population aged 0–10 and 65+ | Population aged 0–10 and 65+ within the area |

In absence of detailed information from official sources, data for the variables considered in Table 3 have been extracted from OpenStreetMap (OSM). We recognize that volunteered geographic information can have gaps in the reliability, quality, and homogeneity of the information due to user participation. However, we consider OSM as a valuable source of data that makes the exposure assessment transferable and applicable in contexts where there is no official geographical information available. Data on the distribution of the population aged 0–10 and 65+ have been instead collected from the Italian National Institute of Statistics (ISTAT, 2011).

*2.6. Combining Vulnerability and Risk with Land Use Information*

At this stage, it is possible to represent the vulnerability map and risk map at the district level. However, the adaptation process in urban environments demands the design of interventions in complex built-up areas, in which public and private spaces, different urban forms and urban functions coexist. Therefore, not all the solutions to mitigate the local impact of heat waves are feasible in all the circumstances, but they must be tailored according to the characteristics of each single area. For this reason, the different elements that compose the city must be identified and acknowledged before the planning and implementation of any adaptation measures.

Land use information, for example, gives important insights about the intraurban differences of a city. The interpolation between vulnerability and risk values (mapped in the hexagonal grid) with land use information allows us to identify the degree of vulnerability and risk for each land use entity. This kind of information guides the planning and design process from the very beginning, since, for example, high vulnerability in the city center or in an industrial area would require very different measures. Land use categories used in this methodology are city center, high and medium residential, low residential, industrial, commercial, and public services. The process aims at facilitating the design of site-specific solutions and identifying the most appropriate planning instrument for each circumstance.

The entire methodology has been tested, besides the case of Padova, in the cities of Milan and Reggio Emilia in collaboration with the local administrations. This gave positive feedback about the soundness of the methodology and its replicability in different contexts.

## 3. Results

The above methodology was applied to the urban fabric of the city of Padova. In this section, results of the analysis are presented both in numerical form and then represented in thematic maps. Results of the sensitivity, adaptive capacity, vulnerability, exposure, and risk assessment have been classified in four categories going from the lowest to the highest values. Table 4 shows that roughly 40% of the urban fabric of Padova is included in the third and fourth range values of sensitivity, revealing high susceptibility to accumulate heat and thereby with the potential to worsening the effects of heat waves.

**Table 4.** Range value of sensitivity.

| Category | Range Value | $m^2$ | % |
|---|---|---|---|
| 1 | 0–0.25 | 5,530,104.15 | 9.54 |
| 2 | 0.25–0.40 | 28,634,644.18 | 49.39 |
| 3 | 0.40–0.50 | 16,783,869.75 | 28.95 |
| 4 | 0.50–1 | 7,016,341.093 | 12.104 |

On the other hand (see Table 5), about 96% of the analyzed area has low or zero value of adaptive capacity. This means that there is low presence of vegetation to mitigate the effects of heat waves.

**Table 5.** Range value of adaptive capacity.

| Category | Value | $m^2$ | % |
|---|---|---|---|
| 1 | 0–0.25 | 37,928,408.50 | 65.43 |
| 2 | 0.25–0.40 | 17,747,418.75 | 30.61 |
| 3 | 0.40–0.50 | 1,728,362.491 | 2.98 |
| 4 | 0.50–1 | 560,769.428 | 0.96 |

The results of the vulnerability assessment, which take into account sensitivity and adaptive capacity values, have also been classified into four categories ranging from 1 (low vulnerability) to 4

(high vulnerability). As can be seen in Table 6, more than 60% of the built-up territory in the city of Padova reveals high level of vulnerability to heat waves.

**Table 6.** Range value of vulnerability to heat waves.

| Category | Range Value | m$^2$ | % |
|---|---|---|---|
| 1 | −1–0 | 10,200,807.21 | 17.59 |
| 2 | 0–0.1 | 10,835,466.65 | 18.69 |
| 3 | 0.1–0.25 | 19,590,159.15 | 33.79 |
| 4 | 0.25–1 | 17,338,526.14 | 29.91 |

Results for the exposure assessment are summarized in Table 7, and they show that the variables considered in the analysis are largely present in about 20% of the territory.

**Table 7.** Range value of exposure.

| Category | Range Value | m$^2$ | % |
|---|---|---|---|
| 1 | 0–0.022 | 25,299,759.5 | 43.64 |
| 2 | 0.022–0.057 | 21,094,054.21 | 36.39 |
| 3 | 0.057–0.11 | 10,608,552.99 | 18.30 |
| 4 | 0.11–1 | 962,592.473 | 1.66 |

Finally, the results of the risk assessment have also been classified in four categories from 1 (low risk) to 4 (high risk). The results in Table 8 show that there is a large risk to be negatively affected by heat waves in about 17% of the city territory, while most of the urban fabric does not present alarming risk values.

**Table 8.** Range value of exposure.

| Category | Range Value | m$^2$ | % |
|---|---|---|---|
| 1 | −1–0 | 10,594,364.976 | 18.2771887 |
| 2 | 0–0.015 | 36,770,419.58 | 63.4355999 |
| 3 | 0.015–0.03 | 6,728,772.703 | 11.6083455 |
| 4 | 0.03–0.133 | 3,871,401.913 | 6.67886594 |

Overall data on the level of vulnerability and risk, albeit significant, do not give any insights about where the most vulnerable and most expose areas of the city are located. Nor this data gives information about the typology of urban fabric that is vulnerable or mostly exposed to heat waves. The results of the analysis have then been represented in thematic maps by making use of the hexagonal grid. Figure 2 shows the geographical dimension of sensitivity, adaptive capacity, vulnerability, exposure, and risk assessment results and highlights the areas of the city where interventions are most urgently needed. The geographical representation of the indices allows us to draw more detailed conclusions. For example, vulnerable areas are not homogeneously spread over the entire city but concentrate in the center and the northeastern part of the city. Moreover, it is visible how exposure values tend to increase while moving closer to the city center, which also shows the highest risk values.

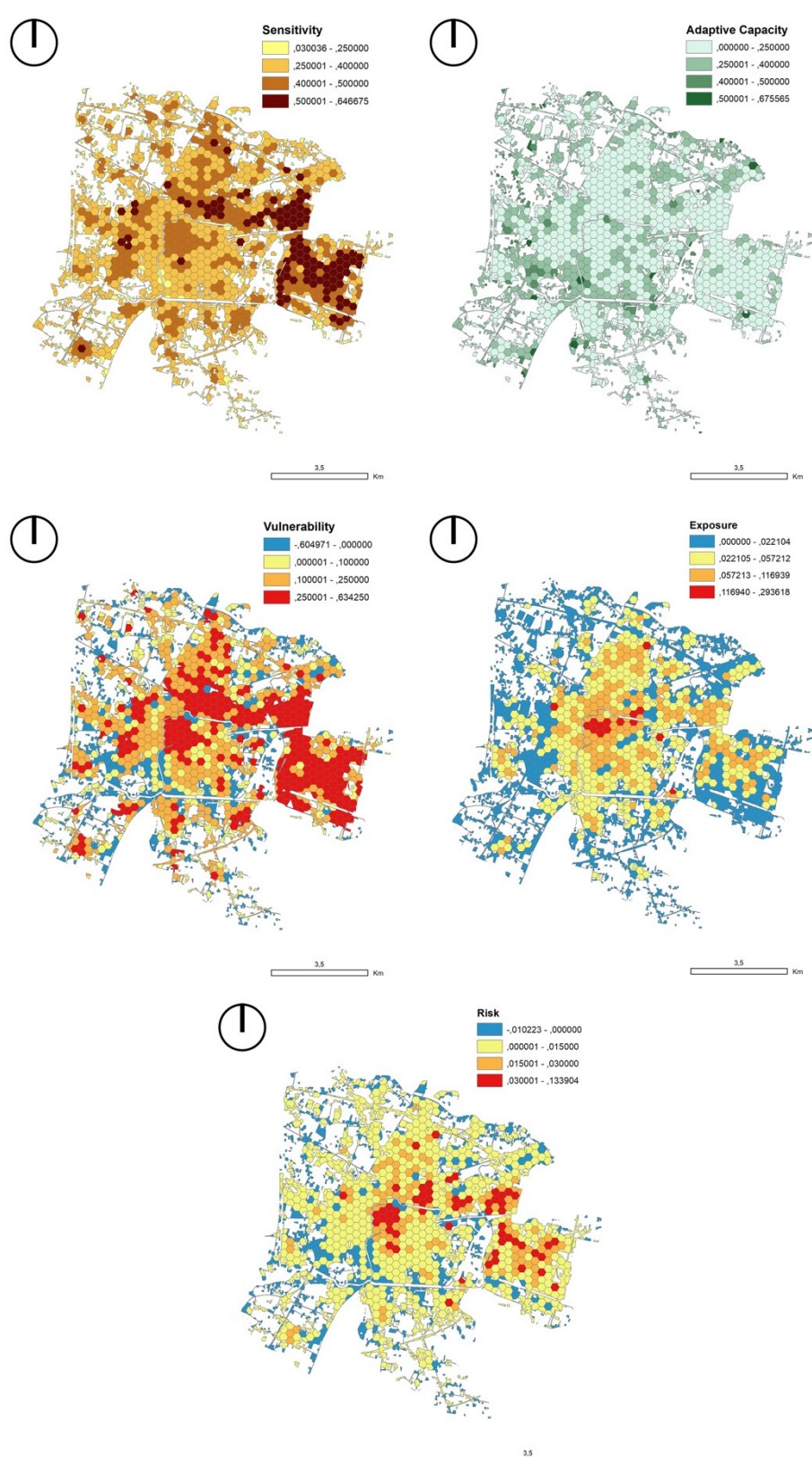

**Figure 2.** The figure shows the maps of the different indicators obtained: sensitivity, adaptive capacity, vulnerability, exposure, and risk to heat waves.

An extra layer of information is provided by the interpolation of vulnerability and risk values with land use entities (Figure 3). In this regard, Table 9 shows how much of each land use category

overlaps with the highest values of vulnerability (categories 3 and 4). The most vulnerable areas are the city center (in full), industrial areas (which are mainly concentrated in the east side of the city), and commercial areas. Parcels of the city occupied by public services and high and medium residential plots are both in the range of high vulnerability values for more than 60%. However, high and medium residential areas cover a much bigger area if compared to public services. Low residential zones have lower vulnerability values and occupy a smaller section of the city.

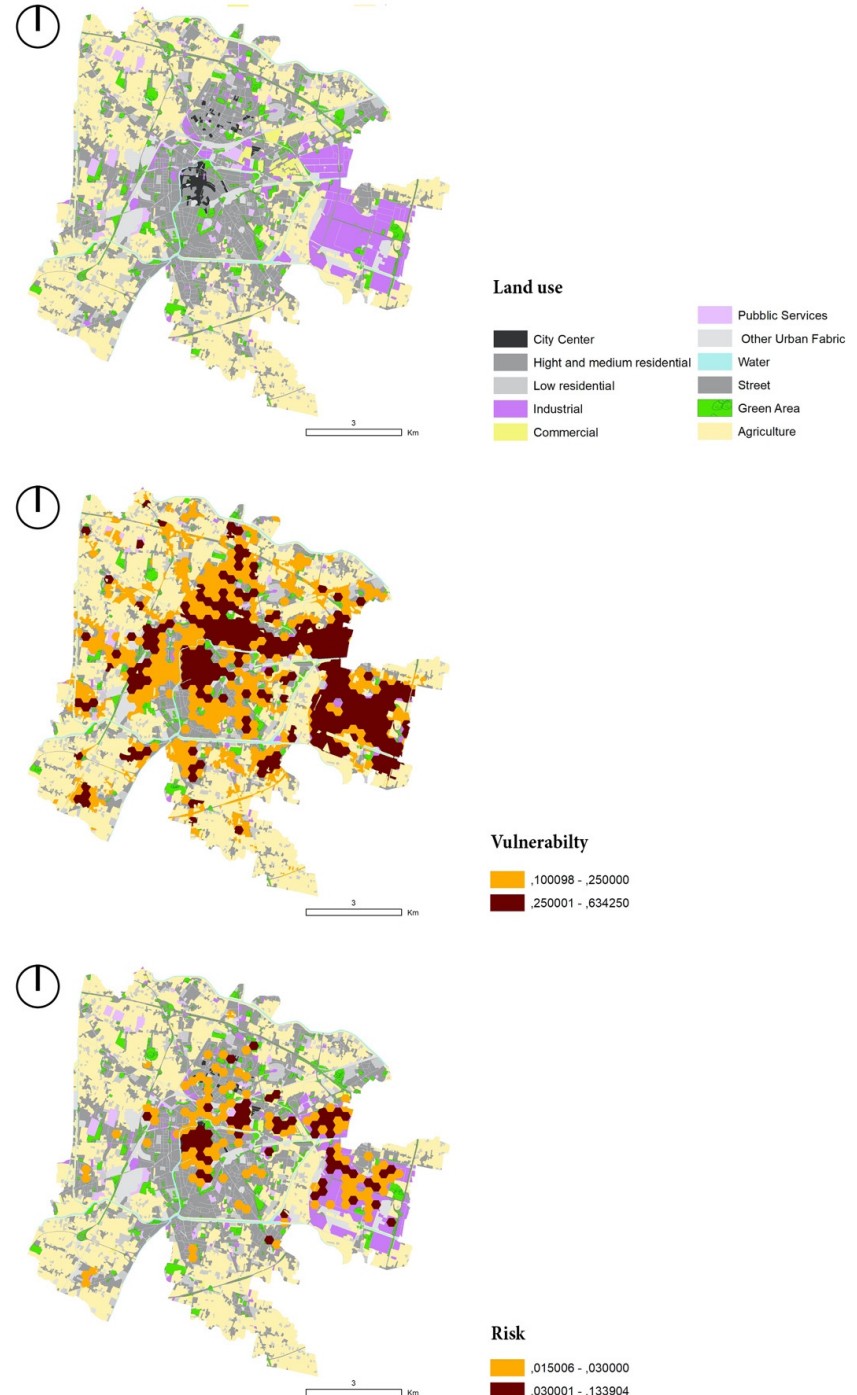

**Figure 3.** The figure shows the maps of the vulnerability and risk assessment for the third and fourth range values.

**Table 9.** Square meters and percentage of vulnerable land for each land use class.

| | Total Area m$^2$ | % of the Built-up Area | Area in the 3 and 4 Ranges of Vulnerability m$^2$ | % of the Vulnerable Area |
|---|---|---|---|---|
| City center | 911,391.14 | 1.57 | 911,391.14 | 100 |
| High and medium residential | 23,991.327 | 41.38 | 15,219,038.20 | 63.43 |
| Low residential | 6,446,979.17 | 11.12 | 1,687,090.08 | 26.16 |
| industrial | 8,156,969.11 | 14.07 | 7,521,374.56 | 92.20 |
| Commercial | 943,328.37 | 1.62 | 862,323.44 | 91.41 |
| Public Services | 1,804,205.48 | 3.11 | 1,230,225.06 | 68.18 |

In a similar way, Table 10 shows the interpolation between areas with high risk values (categories 3–4) with the land use entities. The city center, despite representing less than 2% of the built-up area, is to be considered entirely as a high-risk area. Industrial and commercial areas, albeit to a lesser extent, are also in a potentially risky situation. High and medium residential areas are only 12.5% in a high-risk situation, but it should be noted that this represents 7% of the urban territory of Padova, a much larger area than the city center. Public services areas, despite being largely interested by high vulnerability values, are only in high-risk areas for 14% and cover a very small part of the city.

**Table 10.** Square meters and percentage of risk land for each land use class.

| | Total Area m$^2$ | % of the Built-up Area | Area in the 3 and 4 Ranges of Risk m$^2$ | % of the Risk Area |
|---|---|---|---|---|
| City center | 911,391.14 | 1.57 | 911,391.14 | 100 |
| High and medium residential | 23,991.327 | 41.38 | 3,001,475.12 | 12.51 |
| Low residential | 6,446,979.17 | 11.12 | 93,744.24 | 1.45 |
| industrial | 8,156,969.11 | 1.,07 | 3,925,433.47 | 48.12 |
| Commercial | 943,328.37 | 1.62 | 514,696.87 | 54.56 |
| Public Services | 1,804,205.48 | 3.11 | 256,217.69 | 14.20 |

In addition to identifying where the areas that urgently require intervention are located, the results can also be presented in such a way that it is possible to determine what are the indicators that affect vulnerability or risk values the most. This helps to promptly define the best solutions by tackling the specific weaknesses of the territory. Figure 4 shows two different urban areas, one in the city center (1) and the other in the industrial area (2), both clipped in a hexagonal section. Two levels of information are associated to each hexagon. On the left side, the values of sensitivity, adaptive capacity, vulnerability, exposure, and risk are shown for each area. Moreover, the values of the specific indicators used are displayed on the right side of Figure 4.

By comparing the values of the two areas, it is possible to see that vulnerability levels are higher in the industrial area, whereas the risk levels are the same for the city center and the industrial area. Data on the left side give some first insights into what is mainly determining vulnerability and risk values. However, it is the specific indicators on the right side of the picture that show what are the local characteristics that mainly affect vulnerability and risk values.

Comparing the two sample areas by looking at the most specific factors provides more detailed information. For example, despite a lower building density, the industrial area has higher values in all the other indicators related to sensitivity (see for example the sky view factor values). In addition to this, both areas have few trees and green areas resulting in low adaptive capacity. Furthermore, the elements that are exposed in the two areas are very different, which also results in different risk values. In the industrial area, there is a high concentration of industrial facilities, which are the main elements that are exposed in that area. On the other hand, restaurants, bars, and the resident population are the main factors that result in raised exposure levels in the sample area of the city center.

We argue that the possibility to link vulnerability and risk values to specific characteristics of the city, and the capability of doing so with great detail, not only improves the quality of vulnerability and

risk assessment but also provides a first guidance to urban planners and designers about how and where to intervene.

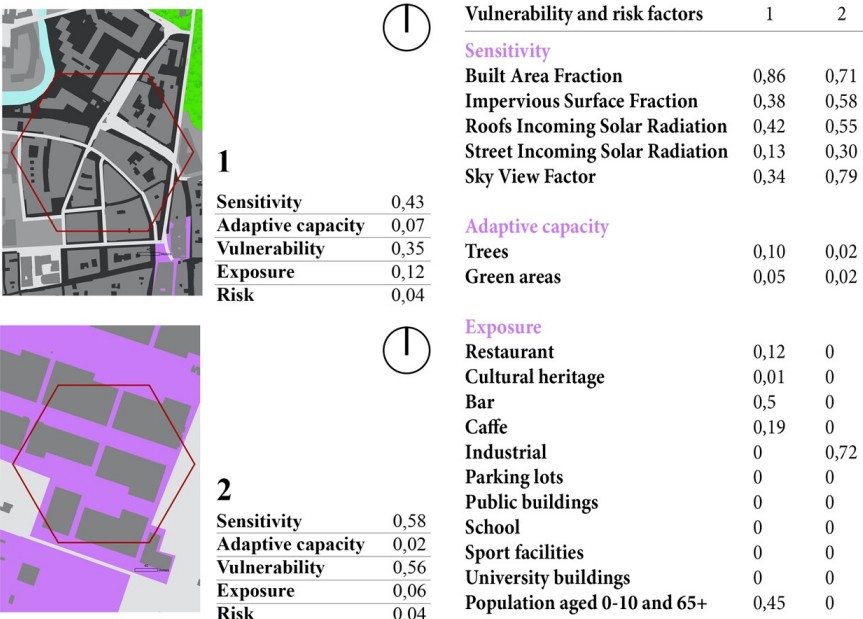

| Vulnerability and risk factors | 1 | 2 |
|---|---|---|
| **Sensitivity** | | |
| Built Area Fraction | 0,86 | 0,71 |
| Impervious Surface Fraction | 0,38 | 0,58 |
| Roofs Incoming Solar Radiation | 0,42 | 0,55 |
| Street Incoming Solar Radiation | 0,13 | 0,30 |
| Sky View Factor | 0,34 | 0,79 |
| | | |
| **Adaptive capacity** | | |
| Trees | 0,10 | 0,02 |
| Green areas | 0,05 | 0,02 |
| | | |
| **Exposure** | | |
| Restaurant | 0,12 | 0 |
| Cultural heritage | 0,01 | 0 |
| Bar | 0,5 | 0 |
| Caffe | 0,19 | 0 |
| Industrial | 0 | 0,72 |
| Parking lots | 0 | 0 |
| Public buildings | 0 | 0 |
| School | 0 | 0 |
| Sport facilities | 0 | 0 |
| University buildings | 0 | 0 |
| Population aged 0-10 and 65+ | 0,45 | 0 |

**1**

| | |
|---|---|
| Sensitivity | 0,43 |
| Adaptive capacity | 0,07 |
| Vulnerability | 0,35 |
| Exposure | 0,12 |
| Risk | 0,04 |

**2**

| | |
|---|---|
| Sensitivity | 0,58 |
| Adaptive capacity | 0,02 |
| Vulnerability | 0,56 |
| Exposure | 0,06 |
| Risk | 0,04 |

**Figure 4.** The figure shows the range value of each indicators for two different hexagons.

## 4. Discussion

The proposed methodology aims to provide the information that is so much needed in the planning and design of adaptation strategies. Information can have different level of detail, and so it can be used for different purposes. For example, the general results of vulnerability and risk assessment presented in a numerical form reveals that around 60% of the urban fabric of Padova has high values of vulnerability to heat waves and 17% of the urbanized territory records high values of risk. This suggests that the physical characteristics of Padova make the city generally vulnerable to heat wave and susceptible to accumulate heat, while the general lower value of exposure mitigates the risk consequently. This kind of result gives an overview of the situation of the city as a whole and can be a wakeup call for taking action. It is also an interesting result for comparative studies, in case the methodology is applied to other contexts.

Furthermore, the representation of the results in thematic maps is a fundamental added-value to the analysis in three ways: (i) vulnerability and risk values are spatialized, adding a geographical dimension to the results and showing that to different parts of the city correspond different levels of vulnerability and risk; (ii) it drastically improves the communication and understanding of the results for a wider group of stakeholders; and (iii) it provides an information layer that can be matched with other georeferenced data.

However, up to this point, the results give information about where it is relevant to implement adaptation strategies in the city, but there are no sufficient details to support planning and designing actions. In this sense, the interpolation of vulnerability and risk values with land use entities is fundamental to move beyond the analytical perspective and to take a more solution-oriented approach. In the case of Padova, there are evidence that, albeit to very different extents, high levels of vulnerability and risk to heat waves can be found across the entire city and in all land use conditions. This is important to start thinking about differentiate solutions according to the typology of the urban fabric. In fact, dealing with high vulnerability and risk values in an historical city center, in a low-density residential zone or in an industrial area, we need to think about different adaptation measures. Different urban planning instruments (e.g., building code, master plan, land use regulation, forestry regulation,

heritage management plans, etc.) might also be required, depending on the area in question and the measure selected.

Furthermore, the possibility to identify which indicators contribute the most to high vulnerability and risk values is very useful to steer the planning and design of the most effective adaptation measures. For example, data presented in Figure 4 suggest that adaptation measures in the industrial area must focus on increasing trees and green areas while reducing impervious surfaces. Other variables that contribute to increase vulnerability, such as the SVF, are difficult to change in existing built-up areas and might be considered in new industrial development. On the other hand, greening measures and reducing impervious surfaces might not be enough to adapt the city center to heat waves. In fact, the presence of many urban functions in this part of the city might require solutions other than physical transformations of the urban fabric. This includes, for example, the relocation of people and urban functions, the establishment of early warning systems, or financial tools for risk management.

We argue that the vulnerability and risk assessment as it is proposed in this paper differs from previous similar studies (e.g., references [23,27]), mainly because it does not show only what is vulnerable or at risk, but it clearly points out what are the factors that increase vulnerability and risk, and it does so by including in the analysis properties of surface structure and surface cover at a very detailed scale.

Considering the multiple levels of detail of the information produced through the vulnerability and risk assessment, we argue that the methodology can contributes to the planning process on multiple fronts: (i) primarily in the preparatory/analysis phase by increasing the quantity/quality of the information available for a certain territory; (ii) in the development of strategies and measures, giving first guidelines on where and how to do adaptation; (iii) in the implementation phase, although to a lesser extent, to adjust adaptation projects under specific circumstances; and (iv) in the monitoring phase, considering that the geo-database and the indicators can be updated showing changes over time and revealing the effects of the implementation of adaptation strategies and measures.

As regards the methodology, it is important to highlight a few points. The hexagonal grid showed to be a powerful analytical tool with a great potential to support climate-proof planning in different ways. First, it gives a geographic dimension to vulnerability and risk values, and in this way, it makes it possible to identify where there is more urgent need for action. The hexagonal grid enables the integration between different kind of information, laying the basis for an integrated planning process. Second, the geodatabase and its graphic representation in the shape of hexagonal grid collect and turn vector information into attributes of the hexagon. The management of data is therefore simplified, making it possible to interpolate a large amount of information. Third, the methodology can function as a monitoring tool. In fact, the analysis gives a picture of the current situation of the territory composed by the value of each indicator. Adaptation measures implemented on the territory supposedly change the values of the indicators so that it is possible to monitor the progresses.

The exposure assessment has proved to require particular attention as it implies that information about population and urban functions (or any other object considered to be exposed) is available in a uniform manner on the entire analyzed territory. However, this kind of information is often not readily available in a standard format. In the case of Padova, for example, a large part of the information needed for the assessment was not available in the geodatabase of the public administration. The problem has been overcome by processing information obtained from Open Street Maps. In order to complete the assessment, OSM was an essential source of information, and it makes the methodology replicable in other contexts. However, it might not always be possible to guarantee that the information and the quality of the information are homogeneous on the entire area. Efforts by the public administrations (or other institutions) to increase the quantity and quality of geographical data and systematically collect, organize and update information will result in more reliable vulnerability and risk assessments.

It is important to be aware that the methodology is not meant to be a Decision Support System. The objective here is to provide the best possible information to support the planning and design of adaptation strategies and measures. However, understanding how the information is used and

how the decision-making process is conducted was not the objective of this work and it should be considered for future research. In fact, we assume that the development of adaptation strategies also depends on political, social and economic factors that play a great role in decision-making and that have not been considered in our methodology.

## 5. Conclusions

There is general consensus about the urgency of taking action to counteract climate change at the city level, but although some progress has been made in the field of mitigation, the planning and implementation of adaptation measures at the local level is advancing slowly. The adaptation of cities to climate change requires taking action on several fronts and a substantial revision of how they are planned, designed, and managed. However, the lack of data and information to support the planning process is often mentioned as a factor that is hampering the adaptation processes at the local level. In this paper, we developed and tested a methodology for heat stress vulnerability and risk assessment to support designers, planners and decision makers to develop adaptation strategies and measures at the local level regardless the availability of reliable high-resolution climate data. We adapted the IPCC methodology for vulnerability and risk assessment to be used at the district level by combining high-resolution spatial information and crowdsourcing geospatial data to develop sensitivity, adaptive capacity, vulnerability, exposure, and risk indicators. The methodology was then tested on the urban fabric of the city of Padova, Italy.

The methodology, which has been applied to other Italian contexts besides the case of Padova, proved to be very useful to quantify and identify the location of the most vulnerable and the most at-risk areas when considering the potential impacts of heat waves. More than that, combining vulnerability and risk indicators with land use information allowed moving from the analysis stage to the initial phase of the planning process and to start thinking about the adaptation measures that are best suited for each typology of urban area.

Nevertheless, we recognize some limitations. It is important to note that, in the application presented in the paper, the indicators used have all the same weight. Changing the weight of the indicators might drastically affect the results of the vulnerability and risk assessment. We consider this as particularly relevant for the exposure calculation. The decision of giving greater importance, and therefore greater weight, to some elements (e.g., schools rather than industrial facilities) is a political choice and it should be addressed and properly discussed. It is also important to emphasize that the indicators used have been considered in relation to heat waves. Although the process of analysis can be maintained, the indicators should be adjusted according to the hazard considered. The methodology was designed to be replicable in different contexts, however we recognize that the availability of high-resolution geographical information determine the quality of the analysis. In this sense, New Technologies (NT), and particularly Information and Communication Technologies (ICT), give a great contribution to supply the information required.

In order to improve it, and verify its transferability in different contexts, the same methodology is going to be applied, in close cooperation with local administrations, to the territories of the Metropolitan city of Venice. A refinement of the methodology is expected, particularly regarding the definition of the exposure index, since it will be possible to cross different information drawn from shared databases.

In conclusion, considering the needs of urban designers, planners, and local decision makers, we have reasons to consider the methodology as a useful tool to support vulnerability and risk assessment and to boost the development of adaptation strategies at the district level. We also argue that the fact of the methodology has been tested in real life contexts with the endorsement of practitioners from the local administrations also contributes to validate the work presented in this paper.

**Author Contributions:** Conceptualization, D.M.; methodology, D.M.; software, D.M.; validation, D.M. and M.D.F.; formal analysis, D.M.; investigation, D.M. and M.D.F.; resources, D.M. and M.D.F.; data curation, D.M.;

writing—original draft preparation, D.M. and M.D.F.; writing—review and editing, M.D.F.; supervision, F.M.; project administration D.M. and F.M.; funding acquisition, F.M.

**Funding:** This research was funded by LIFE Veneto ADAPT—Central VENETO Cities netWorking for ADAPTation to Climate Change in a multi-level regional perspective (LIFE16 CCA/IT/000090).

**Conflicts of Interest:** The authors declare no conflict of interest. The funders had no role in the design of the study; in the collection, analyses, or interpretation of data; in the writing of the manuscript, or in the decision to publish the results.

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
