# Peer review of "Mapping Heat Stress Vulnerability and Risk Assessment at the Neighborhood Scale to Drive Urban Adaptation Planning"

_sustainability, doi:10.3390/su12031056_

Round 1
Reviewer 1 Report
This is a very interesting study. While the manuscript is well organized, there are still some issues need to be addressed before accepting for publication. Please find my comments below.
An abstract typical should include the main ideas of each section of the article and significant results.
Introduction. The contribution of this study has not been highlighted. Therefore, a revision is needed for this section.
Methodology. I highly suggest the authors provide a flow chart of the proposed study, it will give readers a clear overview of your study, and also will help readers to follow your research.
Different numbers of variables have been used for calculating vulnerability and risk. How to decide which variable should be included in the calculation? Only two adaptive capacity indices, but 11 exposure indices are included in this study, does it reasonable?
Results section. Several tables were included in this section. The authors just simply explain the results included in the tables, much deeper explanation and discussion are needed.
Discussion. Compare the proposed study with previous studies are needed in this section.
Conclusions. Limitations of this study and possible future research directions are also needed in this section.
Minor issues:
Line 37, [3]; [4]; [5]; [6] should be included in the manuscript as [3-6]. Please double-check the journal’s format requirement throughout the whole manuscript. Line 154 please include a reference for the formula you provided in the manuscript. Line 182, same as Line 154 Text on almost all the figures is too small, the north arrow also needed for each figure. The reference format needs to be revised based on the journal’s requirement. The whole name needs to be given in the manuscript, then you can use the abbreviation later in the manuscript. For instance, IPCC.
Author Response
Dear editors and reviewer,
We are pleased to submit the new version of our manuscript. We would like to thank the reviewer for their detailed and useful comments and feedbacks, which really helped us to improve the manuscript.
You can find below a detailed explanation of our changes and improvements to the article according to the comments and suggestions.
---
1
Comment: An abstract typical should include the main ideas of each section of the article and significant results.
Response: Abstract has been revised.
2
Comment: Introduction. The contribution of this study has not been highlighted. Therefore, a revision is needed for this section.
Response: The contribution of this study is better explained in the introduction of this new version. This study aims to develop a methodology to perform vulnerability and risk assessment at the neighborhood scale, also taking into account properties of surface structure and surface cover into the analysis. In doing so we contribute to the research field on climate change adaptation, vulnerability and risk analysis in urban areas.
3
Comment: Methodology. I highly suggest the authors provide a flow chart of the proposed study, it will give readers a clear overview of your study, and also will help readers to follow your research.
Response: A flow chart has been included in the methodology section of this new version.
4
Comment: Different numbers of variables have been used for calculating vulnerability and risk. How to decide which variable should be included in the calculation? Only two adaptive capacity indices, but 11 exposure indices are included in this study, does it reasonable?
Response: The objective was to provide a methodology ready to be used. Therefore the typology of variables were is backed by the literature, while the number of variables depended on the availability of data, that is fundamental for the actual operationalization of the methodology. The different number of indices between adaptive capacity and exposure is not to be considered problematic as the values have been normalized according to the number of variables available.
5
Comment: Results section. Several tables were included in this section. The authors just simply explain the results included in the tables, much deeper explanation and discussion are needed.
Response: Results section has been increased with deeper explanation as suggested, although we still prefer not to expand too much this section in order to maintain a neat structure of the results. Relevant issues to be discussed are then included in the discussion section.
6
Comment: Discussion. Compare the proposed study with previous studies are needed in this section
Response: A detailed comparison with previous studies was not an objective of our paper, however we understand the suggestion and we included some comments on this issue in the discussion section of this new version.
7
Comment: Conclusions. Limitations of this study and possible future research directions are also needed in this section.
Response: Limitations of the study are included in the conclusion. Future research directions are also now included in the conclusion section, in particular with reference to the application of the methodology in the Metropolitan City of Venice.
8
Comment: Line 37, [3]; [4]; [5]; [6] should be included in the manuscript as [3-6].
Response: References have been revised.
9
Comment: Line 154 please include a reference for the formula you provided in the manuscript. Line 182, same as Line 154.
Response: References have been included
10
Comment: Text on almost all the figures is too small, the north arrow also needed for each figure
Response: Figures have been revised in this new version of the paper.
11
Comment: The reference format needs to be revised based on the journal’s requirement.
Response: The reference has been revised
12
Comment: The whole name needs to be given in the manuscript, then you can use the abbreviation later in the manuscript. For instance, IPCC.
Response: The whole name is given in this new version of the paper.
Reviewer 2 Report
I have certainly enjoy and learn reading your paper, congratulations.
The paper is well written, rellevant, and presents certainly an interesting advance into characterising heterogeneity withing urban areas and therefore helping specifying concrete responses.
There is therefore little I can say but still some minor issues could be re-think about:
First, the title could include "heat stress vulnerability and risk analysis" because this is really what the paper is about, and is certainly more informative.
Second, in page 12 you mention twice Figure 4 but you seem to mean Figure 3.
Third, in the same page, line 277 you mention "the city centre has higher risk level" but data in table do not support (is 0,04 for both).
Four, I found out that the two different urban areas are 1- for city centre and 2 for industrial area, but you could write it and not make readers to guess ; - )
Otherwise the paper is ready in my opinion.
Best,
Author Response
Dear editors and reviewer,
We are pleased to submit the new version of our manuscript. We would like to thank the reviewer for their detailed and useful comments and feedbacks, which really helped us to improve the manuscript.
You can find below a detailed explanation of our changes and improvements to the article according to the comments and suggestions.
1
Comment: First, the title could include "heat stress vulnerability and risk analysis" because this is really what the paper is about, and is certainly more informative.
Response: We agree with the suggestion, and we changed the title accordingly.
2
Comment: Second, in page 12 you mention twice Figure 4 but you seem to mean Figure 3.
Response: Figure numbers have been revised.
3
Comment: Third, in the same page, line 277 you mention "the city centre has higher risk level" but data in table do not support (is 0,04 for both).
Response: The sentence has been corrected in this new version of the paper.
4
Comment: Four, I found out that the two different urban areas are 1- for city centre and 2 for industrial area, but you could write it and not make readers to guess
Response: The definition of the areas has been specified in this version.
Reviewer 3 Report
The authors have used the already developed IPPC methodology for vulnerability and risk analysis.
In this article, they have presented their intention to support designers, planners and decision makers to develop and operatively implement adaptation strategies and measures at the local level, i.e. for city of Padova.
But how exactly can the results of their research (calculation + maps) help planners to plan adaptation strategies for Padova city? I find this as a main weakness of the article. What should be the first steps of planning adaptation process. It would be very useful if authors can put calculations + maps into the planning context of Padova city (waht is a present mechanism for land use planning in Padova? In which step of land use planning procedure this should be done?, etc...)
I would suggest:
to put a new subtitle in methodology part informing the reader about the present situation on heat islands in city of Padova and about the general methodology of land use planning in Padova to re-write discussion and conclusions in a way that a reader can have a better picture about the utilisation of author´s research results for better land use planning of Padova city and Padova city neighborhoods. At present form, discussion and conclusions are just about the applied methodology.
Author Response
Dear editors and reviewer,
We are pleased to submit the new version of our manuscript. We would like to thank the reviewer for their detailed and useful comments and feedbacks, which really helped us to improve the manuscript.
You can find below a detailed explanation of our changes and improvements to the article according to the comments and suggestions.
1
Comment: But how exactly can the results of their research (calculation + maps) help planners to plan adaptation strategies for Padova city? I find this as a main weakness of the article. What should be the first steps of planning adaptation process. It would be very useful if authors can put calculations + maps into the planning context of Padova city (what is a present mechanism for land use planning in Padova? In which step of land use planning procedure this should be done?, etc...
Response: We tried to cover this issue in the introduction, the methodology and in the discussion of this new version. However, we do not refer specifically to the planning context of Padova, but to a more overarching planning process. This is because we want to make clear that the methodology can be applied in different contexts and not only in Padova.
2
Comment: I would suggest:
to put a new subtitle in methodology part informing the reader about the present situation on heat islands in city of Padova and about the general methodology of land use planning in Padova to re-write discussion and conclusions in a way that a reader can have a better picture about the utilisation of author´s research results for better land use planning of Padova city and Padova city neighborhoods. At present form, discussion and conclusions are just about the applied methodology.
Response: A new subtitle has been included in the methodology section. And discussion and conclusions have been revised.
Reviewer 4 Report
The paper “Mapping risk and vulnerability assessment at the neighborhood scale to drive urban adaptation planning” is very interesting and addresses a topic which evoke the interest of the main media all over the world.
The paper is very clear and complete. Also the results and discussion are well developed and described in appealing way.
For the sake of clarity I suggest to authors to include in paragraph two a flowchart able to synthetize the whole procedure.
In conclusion I suggest to the authors to integrate the paper adding the flowchart in paragraph two.
Author Response
Dear editors and reviewer,
We are pleased to submit the new version of our manuscript. We would like to thank the reviewer for their detailed and useful comments and feedbacks, which really helped us to improve the manuscript.
You can find below a detailed explanation of our changes and improvements to the article according to the comments and suggestions.
1
Comment: For the sake of clarity I suggest to authors to include in paragraph two a flowchart able to synthetize the whole procedure.
Response: A flow chart has been included in the methodology section of this new version.
Round 2
Reviewer 1 Report
The manuscript has been significantly improved by the authors, it should be good enough to be accepted for publication now.
Reviewer 3 Report
I appreciate all revisions that have been done by authors